# Bioactive Triterpenes of *Protium heptaphyllum* Gum Resin Extract Display Cholesterol-Lowering Potential

**DOI:** 10.3390/ijms22052664

**Published:** 2021-03-06

**Authors:** Giuseppe Mannino, Piera Iovino, Antonino Lauria, Tullio Genova, Alberto Asteggiano, Monica Notarbartolo, Alessandra Porcu, Graziella Serio, Giorgia Chinigò, Andrea Occhipinti, Andrea Capuzzo, Claudio Medana, Luca Munaron, Carla Gentile

**Affiliations:** 1Department of Biological, Chemical and Pharmaceutical Sciences and Technologies (STEBICEF), University of Palermo, 90128 Palermo, Italy; giuseppe.mannino@unipa.it (G.M.); antonino.lauria@unipa.it (A.L.); monica.notarbartolo@unipa.it (M.N.); graziella.serio01@unipa.it (G.S.); 2Biosfered S.R.L., 10148 Turin, Italy; p.iovino@biosfered.com (P.I.); a.asteggiano@biosfered.com (A.A.); 3Department of Life Sciences and Systems Biology, University of Turin, 10123 Turin, Italy; tullio.genova@unito.it (T.G.); giorgia.chinigo@unito.it (G.C.); luca.munaron@unito.it (L.M.); 4Department of Molecular Biotechnology and Health Sciences, University of Torino, 10125 Torino, Italyclaudio.medana@unito.it (C.M.); 5Abel Nutraceuticals S.R.L., 10148 Turin, Italy; a.porcu@abelnutraceuticals.com (A.P.); a.occhipinti@abelnutraceuticals.com (A.O.); a.capuzzo@abelnutraceuticals.com (A.C.)

**Keywords:** hypercholesterolemia, gene expression, HMGCR, PCSK9, PPARα, enzymatic activity, molecular docking, statin, monacolin, breu branco

## Abstract

Hypercholesterolemia is one of the major causes of cardiovascular disease, the risk of which is further increased if other forms of dyslipidemia occur. Current therapeutic strategies include changes in lifestyle coupled with drug administration. Statins represent the most common therapeutic approach, but they may be insufficient due to the onset of resistance mechanisms and side effects. Consequently, patients with mild hypercholesterolemia prefer the use of food supplements since these are perceived to be safer. Here, we investigate the phytochemical profile and cholesterol-lowering potential of *Protium heptaphyllum* gum resin extract (PHE). Chemical characterization via HPLC-APCI-HRMS^2^ and GC-FID/MS identified 13 compounds mainly belonging to ursane, oleanane, and tirucallane groups. Studies on human hepatocytes have revealed how PHE is able to reduce cholesterol production and regulate the expression of proteins involved in its metabolism. (*HMGCR*, *PCSK9*, *LDLR*, *FXR*, *IDOL*, and *PPAR*). Moreover, measuring the inhibitory activity of PHE against HMGR, moderate inhibition was recorded. Finally, molecular docking studies identified acidic tetra- and pentacyclic triterpenoids as the main compounds responsible for this action. In conclusion, our study demonstrates how PHE may be a useful alternative to contrast hypercholesterolemia, highlighting its potential as a sustainable multitarget natural extract for the nutraceutical industry that is rapidly gaining acceptance as a source of health-promoting compounds.

## 1. Introduction

Hypercholesterolemia is a form of hyperlipidemia that is characterized by the presence of high levels of circulating low-density lipoprotein cholesterol (LDL-C). It is the major cause of arteriosclerosis [1], which may lead to serious cardiovascular diseases (CVDs) such as hypertension, cerebrovascular disease, peripheral arterial disease, coronary disease, deep vein thrombosis, and pulmonary embolism [2]. CVDs are the leading cause of death globally and were responsible for 31% of all global deaths in 2016, as estimated by WHO (World Health Organization) [3]. According to the different hypercholesterolemia risk levels, a number of therapeutic strategies are actually available [1]. As a first step, changes in lifestyle and diet behaviors are strongly recommended [1,4]. In particular, the consumption of foods with high carbohydrates, saturated fats, and high cholesterol levels should be avoided, while increasing the intake of foods rich in fiber, potassium, unsaturated fats, and saponins [5,6]. Statin therapy represents the most common pharmacological approach for the management of hypercholesterolemia, and the current guidelines recommend a treatment with the maximally tolerated dose of statins [1]. This class of drugs exerts cholesterol-lowering effects by affecting the enzymatic activity of 3-hydroxy-3-methyl-glutaryl coenzyme A reductase (HMGCR) and also increasing the hepatic expression of the LDL receptor (LDLR) and reducing low density lipoprotein (LDL) and very low density lipoprotein (VLDL) synthesis [7]. However, in most cases, statin monotherapy was insufficient to achieve the recommended LDL-C level [2,8], also because of the onset of drug-resistance mechanisms [7]. In these cases, statin therapy is replaced or accompanied by the administration of other drugs, such as fibrate, niacin, and ezetimibe [1,8]. Furthermore, recent epidemiological evidence has shown that patients under treatment with statin, alone or in combination with other medicines, might present muscle-related symptoms or other pharmacological side effects, which lead to the discontinuation of treatments [9,10]. For these reasons, patients with a clinical picture of low hypercholesteremic severity prefer applying nonpharmacological approaches, such as phytotherapeutic ones, because they are perceived as natural, safer, and free of side effects [11]. Among the different phytochemical approaches, red yeast rice (RYR) has been used for centuries as herbal medicine in China to alleviate symptoms derived from hypercholesterolemia conditions [12,13,14]. Currently, RYR is also present in the global market as a dietary supplement (DS), and, in 2008, it was estimated that USD 20 million was spent by American consumers exclusively on this DS category [12]. The antihypercholesterolemic activity of RYR is attributed to the presence of monacolin K [7,13] and citrinin [15]. While monacolin K efficacy is linked to its structural equality to lovastatin [16,17], citrinin is a mycotoxin with documented antihypercholesterolemic properties [15,18]. However, recent clinical studies attributed significant side effects to both compounds. In particular, equally to other statins, monacolin K displayed anaphylaxis, hepatotoxicity, central nervous system complaints, rhabdomyolysis effects, and a high possibility of developing diabetes mellitus [7,14,19]. On the other hand, the nephrotoxicity, carcinogenicity, cytotoxicity, and teratogenicity of citrinin are well known even at a very low dosage [20]. For these reasons, in the last few decades, the research of new medicinal plants as an alternative approach for the treatment of hypercholesterolemia is on-going [21,22,23]. In particular, attention is focused on the identification of new phytochemicals that may exert an antihypercholesterolemic action through a different mechanism of action from what has been reported for statins [24].

*Burseraceae* is a pantropical-distributed family consisting of 21 genera and more than 700 species [25], of which 135 belong to the *Protium genus* [25,26]. *Protium heptaphyllum* is a traditional medicinal plant distributed in the Amazon region, especially in Brazil [27]. Equally to other species belonging to the *Burseraceae* family, *P. heptaphyllum* also yields large amounts of resin from the trunk wood [26]. The resin, popularly known as “almécega” or “breu branco”, is collected and used locally in folk medicine practices for its analgesic, anti-inflammatory, and expectorant effects; some recent works have demonstrated several of those activities. In particular, concerning central actions, Aragão and colleagues showed in a murine model system both anticonvulsant effects related to the GABAergic system [28,29], and anxiolytic and antidepressant effects involving benzodiazepine-type receptors and noradrenergic mechanisms [26]. In addition, antinociceptive potential was also demonstrated [30,31]. The traditional use of the oily resin in inflammatory syndromes is justified by data from several in vivo studies. In particular anti-inflammatory effects of *P. heptaphyllium* gum resin extracts were showed in pleuritis [32], gastric ulcer [30,33], carrageenan induced-edema [34] acute periodontitis [35], and pancreatitis [35]. However, the mechanisms involved in the observed anti-inflammatory action were not clarified. More recently, antibacterial activity [36,37,38] and effects on blood sugar level and lipid profile have also been highlighted for the essential oil obtained from *P. heptaphyllum* gum resin [39,40,41]. The observed effects on lipid metabolism involved a decrease of total cholesterol, LDL-C, serum triglycerides, and VLDL, with an elevation of high-density lipoprotein cholesterol (HDL-C). In addition, antilipidemic effects were associated with modulatory actions on the secretion of hormones related to fat and carbohydrate metabolism [40]. In a more recent study, in order to investigate the mechanism involved in the hypolipidemic action of the resin, De Melo et al. evaluated the antiadipogenic activity of an α- and β-amyrin mixture in 3T3-L1 murine adipocytes [42]. They showed significant suppression of adipocyte differentiation by downregulation of adipogenesis-related transcription factors, including PPARγ (Peroxisome Proliferator-Activated Receptor α9 and C/EBPα (CCAAT-enhancer binding protein α) [42].

In the past twenty years, the chemical composition of the essential oil obtained from the resin of *P. heptaphyllum* has been largely investigated. In particular, De Lima [38] and Rüdigera [26] showed as this resin was rich in terpenoid compounds, especially monoterpenes (α- and γ-terpinene, p-cymenol, terpinolene, α-pinene, p-cymene, 3-carene, limonene), sesquiterpenes (g- and d-cadinene), and pentacylic triterpenes belonging the classes of ursane, oleanane, tirucalane, lupane, and taraxtane. Among them, α-amyrin and β-amyrin have always been identified as the main phytochemicals responsible for the observed in vitro and in vivo biological activities.

In this work, we investigate the hypocholesterolemic potential of a new chemically characterized *P. heptaphyllum* resin extract (PHE). With this purpose, we measured the in vitro effects of PHE on cholesterol biosynthesis in comparison with monacolin K [43]. Moreover, we also evaluated if PHE affects HMGCR, evaluating both changes in enzymatic activity and the binding ability of the molecules identified in PHE. Finally, studies on the expression of genes related to cholesterol metabolism were performed.

## 2. Results and Discussion

### 2.1. HPLC-APCI-HRMS^2^(High Performance Liquid Chromatography coupled with Atmospheric Pressure Chemical Ionization and High Resolution Mass Spectrometry) and GC-FID/MS (Gas Chromatography coupled with Flame Ionization Detector and Mass Spectrometry) Analyses identified Tetra- and Pentacyclic Triterpenoic compounds as the main constituent of PHE

The composition of PHE was assessed by gas and liquid chromatographic analyses coupled to mass spectrometry. Eleven compounds were identified (Figure 1) and quantified (Table 1). 

Among them, two were small-sized volatile organic compounds (Δ3-carene (#1); p-cymene (#2)), six were nonacidic pentacyclic triterpenes (α-amyrin (#3); β-amyrin (#4); α-amyron (#5); β-amyron (#6); brein (#7); maniladiol (#8)), two were acid pentacyclic triterpenois (oleanolic acid (#9); ursolic acid (#10)), and three were tetracyclic triterpenoids (elemonic acid (#11); α-elemolic acid (#12); β-elemolic acid (#13)).

The fraction containing the volatile organic compounds (VOCs) was the least representative in PHE, reaching only about 1.5% of total composition. Meanwhile, Compound #1 was a bicyclic monoterpene consisting of fused cyclohexene and cyclopropane rings, and #2 was the alkylbenzenic form of #1, which originated after pyrolysis under oxidative conditions [44]. Both compounds are naturally distributed in several aromatic plants and gum resins [45], and they are widely used as ingredients for the realization of cosmetics, such as perfumes [45].

Nonacidic triterpenoids (NATs) and acidic triterpenoids (ATs) were the most abundant molecules in PHE, representing more than 28% and 39% of total composition, respectively. Among the identified NATs, the compounds having a ursane skeleton (#3, #5, and #7) were more predominant with respect to those composed by an oleanane skeleton (#4, #6, and #8). Compounds #3 and #4 contributed to more than 20% of the PHE composition. They are compounds well-known in the literature to exert various pharmacological actions, including analgesic, anti-inflammatory, gastroprotective, hepatoprotective, and hypolipidemic properties [46]. #5, #6, #7, and #8 are compounds originated by the structural changes of #3 and #4. In particular, #5 and #6 are the oxidized forms, respectively, of #3 and #4; meanwhile, #7 and #8 are their 5-hydroxy forms. However, #5, #6, #7, and #8 contribute, together, just 20% of the identified NATs and only 6% of total identified and quantified compounds in PHE. Concerning ATs, HPLC analysis revealed the presence of two pentacycles (#9 and #10) and three tetracycles (#11, #12, and #13) having at least one acid function. Regarding the ATS with a tetracyclic core, the identified compounds had a very similar structure, belonging to the tirucallane groups and exclusively differing, for the substituent group, in the C_3_ position (Figure 1). Among the ATs having pentacyclic scaffolds, #9 belongs to the oleanane group and #10 has a ursane skeleton. ATs are widely distributed in the plant kingdom, and they are most abundant in particular plant gum resins [47,48,49]. Growing evidence has shown a greater attitude for ATs to bind the active site of different enzymes, thanks to the presence of one or more carboxylic substituents in the chemical scaffolds that are able to accept noncovalent bonds easily [50,51,52,53].

### 2.2. PHE Decreases Total Cholesterol in Hepatocytes

In order to evaluate cholesterol production, hepatocytes were treated for 6 or 12 h with PHE (25–200 µg mL^−1^). The same experimental conditions were used to assay LHA (0.01–10 µg mL^−1^) as positive control (Figure 2). LHA is a molecule originated by in vivo metabolism of lovastatin [54,55]. Indeed, lovastatin is a prodrug presenting a γ-lactone closed ring in the form that it is administered. The closed ring strongly limits its inhibitory action on HMGCR, but it can be easily hydrolyzed in vivo into different active metabolites, which appear in plasma after 24 h after oral administration [55,56]. Among them, the most effective cholesterol-lowering agent is LHA, in which the γ-lactone closed ring is hydrolyzed, forming a β-hydroxy acid function [54,55].

No significant decreases in cholesterol production were observed at the tested concentrations after 6 h of treatment with LHA or PHE (Figure 2A). On the other hand, both LHA and PHE were able to negatively affect the cholesterol amount in a dose-dependent manner upon 12 h incubation (Figure 2B). In particular, LHA lowered the cholesterol levels at all the tested dosages, with a decrement of about 50% at 1 µg mL^−1^ or higher concentrations. Our data are in good agreement with the literature for LHA [55,56,57]. Regarding PHE, treatment with 200 µg of the gum resin extract was able to reduce cholesterol levels in a similar way to 10 µg mL^−1^ of the pure LHA molecule. 

The observed effect on cholesterol after treatment with PHE may be correlated to the peculiar phytochemical composition of the gum resin extract. In particular, chromatographic analysis has shown how PHE is a rich source of triterpenes and triterpenoic acids mainly belonging to ursane (**#3**, **#5**, **#8**, and **#10**), oleanane (#4, **#6**, **#7**, and **#9**), and tirucallane (**#11**, **#12**, and **#13**) classes. Previous studies have evidenced how cyclic triterpenes may exert several beneficial effects in metabolic disorders, modulating factors involved in glucose, lipid, and cholesterol metabolism. For example, Tang and coworkers demonstrated the decrease of cholesterol and fatty acid biosynthesis by the inhibition of SREBP (sterol regulatory element-binding protein) after interaction of SCAP (SREBP cleavage activating protein) with betulin, a cyclic triterpene having a lupane skeleton very similar to the compounds identified in PHE [58].

Concerning the phytochemicals identified and quantified in PHE, #3 and #4 are two pentacyclic triterpenoids whose cholesterol-lowering activity is well known and already reported in the literature [46]. In particular, a recent in vivo study highlighted the preventive role of #3 in attenuating high fructose diet-induced metabolic syndrome [59]. Additionally, #9 and #10 have been reported to exert in vivo cardiovascular, antihyperlipidemic, and antioxidant effects in a Dahl salt-sensitive rat model [60]. Moreover, a potential mechanism of action involving PPAR-α activation, a key role receptor involved in glucose and lipid homeostasis, has been proposed [61]. 

### 2.3. Triterpenoic Acids of PHE Modulate HMGCR Activity Assuming Similar Poses to LHE in the Active Site of the Enzyme

Statins regulate cholesterol levels by acting as reversible and competitive inhibitors of the enzyme HMGCR, which is involved in the conversion of acetyl-CoA into mevalonate, a key step that triggers the biosynthetic process of cholesterol metabolism [62]. This action is due to the structural similarity of activated statins, which have a β-hydroxy-acid function that is mistakenly recognized by the enzyme as its natural ligand (β-hydroxy β-methylglutaryl-CoA) [63,64,65,66]. The experimental evidence on in vitro biosynthesis showed a decrement of total cholesterol after 12 h of hepatocyte treatment with 200 µg mL^−1^ PHE, similar to 10 µg mL^−1^ LHA (Figure 2B). Therefore, we investigated the possibility that phytochemical content in PHE could exert a cholesterol-lowering activity through a statin-like action mechanism and the modulation of HMGCR enzymatic activity. Consequently, HMGR activity was monitored after treatment with LHA (0.01–100 µg mL^−1^) or PHE (10–200 µg mL^−1^) (Figure 3). LHA dampened HMGCR activity in a dose dependent manner. In particular, a complete enzymatic abolition was observed at 100 µg mL^−1^ LHA, with a 50% inhibition between 1–10 µg mL^−1^ LHA. On the other hand, the treatments with PHE exerted a weaker effect, although a dose-dependent inhibition of enzymatic activity was detectable (about 26% reduction at the highest concentration). Kashyap and colleagues, reviewing different patents related to extracts composed of two compounds that we identified in PHE (**#9** and **#10**), have reported inhibitory activity against HMGCR [67], although scientific evidence supporting this hypothesis has not been currently reported in the literature.

Molecular docking studies on HMGCR were performed with the aim of analyzing the binding properties of the identified compounds in PHE (Table 1, Figure 1) and hypothesizing the binding site with the AAs involved. We started in silico studies by selecting both lovastatin and its activated form (LHA) as known HMGCR inhibitors (Table 2). In the past, similar studies have also been performed on phytochemical compounds present in extracts of *Withania coagulans* fruits [68,69].

The position of lovastatin in the binding site of HMGCR is shown as the AAs involved are Val^663^, Ser^684^, Ser^661^, Leu^857^, Cys^561^, Ala^856^, Leu^853^, Asn^755^, Hid^752^, and Lys^692^. However, only Ala^751^, Lys^735^, and Asn^755^ played an important role in binding (Figure 4A). On the other hand, when the AAs involved in LHA–HMGCR complex stabilization were analyzed, significant differences were observed with respect to the lovastatin–HMGCR complex (Figure 4B). In particular, Asp^690^, Ser^664^, Lys^735^, Ala^751^, Lys^692^, Hid^752^, Lys^691^, Glu^559^, Arg^590^, Asn^755^, Leu^857^, Ser^661^, and Ser^684^ surrounded the docked structure, and the complex was further stabilized by the formation of six bonds with some AAs residues (Asp^690^, Ser^684^, Lys^735^, Glu^559^, and Arg^590^). To be precise, these bonds exclusively involved the β-hydroxy-acid moiety that is present only after in vivo activation of lovastatin.

Table 2 also reports the molecular docking results with regard to interactions between HMGCR and several compounds previously identified in PHE (Table 1, Figure 1). Most of the molecules identified in the extract showed low docking score values (IFD score < −1460), indicating a weak interaction with the active site of HMGCR. In particular, #1 and #2 showed low binding affinity to the receptor site. Regarding the compounds having docking scores ranging between −4.5 and −6.0 (#3, #4, #5, #6, #7, #8, and #9), a limited surrounding by AA residues was observed. However, despite #8 recording a docking score equal to −5.356 because of the surrounding of only eight AA residues (Leu^857^, Val^683^, Leu^853^, Arg^590^, Ala^856^, Asp^690^, Lys^691^, and Asn^755^), it had an IFD score comparable to lovastatin, thanks to the significative interactions established with Lys^691^ and Asn^755^ (Figure 5).

On the other hand, among the docked structures, only ATs showed docking scores and IFD values indicative of good interaction with the enzyme (Table 2). Among them, the pentacyclic triterpenoids showed the highest docking scores but the lower IFD values with respect to the tetracycles. In particular, the docked structures of #9 and #10 were surrounded by a large number of AA residues (11 surrounded #9 and 13 surrounded #10). Ten of the observed AA residues surrounded both the molecules (Hid^752^, Lys^691^, Lys^692^, Ser^684^, Asp^690^, Lys^735^, Leu^857^, Arg^590^, Val^683^, Leu^853^), suggesting their important contribution in the stabilizing of both compounds in the active site of HMGCR (Figure 6). Interestingly, nine of them were also found to be involved in the stabilization of LHA inside the binding site (Figure 4B). Concerning the bonds directly involved in the stabilization of the complex, the carboxyl group of the two ATs seems to be as important as in LHA. However, while the acidic function of #9 and #10 made only three bonds (Lys^692^, Ser^684^, and Lys^735^; Figure 6A,C), LHA was further stabilized by three additional bonds (Lys^692^, Ser^684^, Asp^690^, Lys^735^, Leu^857^, and Arg^590^) involving β- and δ-hydroxyl groups (Figure 4B). The differences in binding properties between ATs and LHA may explain the lower docking score values of #9 and #10 with respect to LHA.

Concerning the acidic tetracyclic terpenoids (#11, #12, and #13), despite having low–medium docking scores, they reported the highest IFD scores among the identified compounds in PHE. Moreover, even though they did not reach the IFD values measured for LHA, our data suggest that the three tetracycles were able to better interact with the AA residues present in the HMGR receptor site and, consequently, to exert a comparable in silico inhibitory activity. Differently from the docked structures of #9 and #10, the AAs surrounding #11, #12, and #13 were quite different. In particular, #11 was surrounded by 18 AA residues (Met^657^, Asp^767^, Glu^559^, Gly^560^, Leu^562^, Cys^561^, Ser^852^, Ser^565^, Arg^568^, Ala^856^, Leu^857^, Leu^853^, Arg^590^, Hid^752^, Lys^691^, Asp^690^, Asn^755^, and Met^655^); meanwhile, #13 was surrounded by 16 AAs residues (Ala^751^, Asp^690^, Lys^692^, Ser^684^, Lys^735^, Ser^661^, Val^683^, Leu^857^, Glu^665^, Arg^590^, Met^657^, Lys^691^, Asn^755^, Hid^752^, Leu^853^, and Leu^562^) (Figure 7). Among them, the same AA residues that seem to be important for the stabilization of LHA in the HMGR binding site through interaction with its β-hydroxy acid function (Figure 4B) were also observed in the docked structures of #11 (Arg^590^, Hid^752^, Lys^691^, Asp^690^, Asn^755^, and Met^655^) and #13 (Asp^690^, Ala^751^, Lys^692^, Ser^684^, and Lys^735^). This last aspect may explain the higher values recorded for tetracycles with respect to pentacycles.

### 2.4. PHE Modulates the Expression of Cholesterol-Related Genes 

We demonstrated that cell exposure to 200 μg mL^−1^ PHE for 12 h was able to reduce cholesterol production in a comparable way to 10 μg mL^−1^ lovastatin (Figure 2). On the other hand, results from enzymatic assay showed that 200 µg mL^−1^ of PHE produced about 20% inhibition of HMGCR activity; meanwhile, the same enzyme was inhibited by more than 90% by lovastatin at 10 μg mL^−1^ (Figure 3), as previously reported for statins [70,71]. These results suggested differences in the action mechanism of PHE in reducing cholesterol synthesis with respect to lovastatin. In particular, the effects on cholesterol production may involve mechanisms other than post-translational ones. This hypothesis is also supported by molecular docking studies, in which weaker interactions between the identified compounds in PHE (Table 1, Figure 1) and HMGCR were highlighted when compared with the interaction established with statins (Table 2).

In the past, the ability of plant bioactive compounds to regulate the expression of target genes has been largely demonstrated [72,73]. In particular, it has been reported that gene expression may be modulated by phytochemicals via both genetic and epigenetic mechanisms [74,75]. In particular, phytochemical compounds may affect gene expression via the modulation of several biological targets, such as DNA, lipid rafts, and transcriptional factors, directly binding them or indirectly affecting the cellular redox state [75]. In this work, we investigated whereas the observed effect related to the decrease of cholesterol level after PHE treatment could be linked to changes in gene expression of *HMGCR* via qRT-PCR (quantitative Reverse Transcription Polymerase Chain Reaction) analysis. Cells exposed to PHE (1–50 μg mL^−1^) displayed a dose-dependent downregulation of *HMGCR*. In particular, 10 μg mL^−1^ PHE was able to inhibit 50% *HMGCR* gene expression (Figure 8 and Figure 9). This result is in contrast with the effect reported for statins. Indeed, it is well known in the literature that statins, including lovastatin, interfere with the negative feedback of cholesterol on the transcription of the *HMGCR* gene, resulting in its upregulation [76]. Upregulation of the *HMGCR* gene is also a strong limiting factor for the pharmacological effect of statins [7,76].

In addition to de novo synthesis, cholesterol homeostasis depends on other processes, including cellular uptake, intestinal absorption, and metabolic turnover in biliary salts [77]. With the aim of evaluating if PHE may influence other important processes for cholesterol homeostasis, we evaluated the effects derived by cell exposure to PHE on the gene expression of cholesterol-metabolism-related genes.

LDLRs (low-density lipoprotein receptor) play a key role in regulating LDL-C levels in the blood by influencing the clearance of LDL-C from circulation. The gene expression of the receptor is regulated at both transcriptional and post-translational levels [78]. Meanwhile, transcriptional regulation is dependent on the availability of intracellular cholesterol; post-translational mechanisms depend on the action of PCSK9 (Proprotein Convertase Subtilisin/Kexin type 9) and IDOL (Inducible Degrader of Low-density Lipoprotein Receptor) [78,79]. These two proteins are synthesized in hepatocytes and secreted into plasma. Here, PCSK9 binds LDLR in an extracellular site, interfering with receptor in-membrane recycling, while IDOL marks LDLR with ubiquitin, leading to its degradation by lysosomes [80]. Consequently, high plasmatic levels of PCSK9 and IDOL reduce the amount of LDLR in the hepatocyte membrane, resulting in higher levels of plasmatic LDL-C. Therefore, the discovery of new agents that may inhibit *PCSK9* and *IDOL* expression is a useful tool for the treatment of hypercholesterolemia. This finding is also supported by the role of new anti-PCSK9 monoclonal antibody drugs such as alirocumab and evolocumab [1]. With the aim of evaluating whether PHE could contribute to the regulation of LDLR levels through transcriptional mechanisms, we checked the effects of the treatment on the expression of *LDLR*, *PCSK9*, and *IDOL* genes. Surprisingly, the *LDLR* gene was downregulated (Figure 8). On the other hand, under our experimental conditions, we recorded a reduction in the expression of both *PCSK9* and *IDOL* after treatment with PHE in the tested concentration range (Figure 8). In particular, 80% of *IDOL* and *PCSK9* inhibition was obtained at 50 and 10 μg mL^−1^ PHE, respectively. These results suggest a stabilization of LDLR through post-translational mechanisms.

Since high levels of intracellular cholesterol induce the inhibition of the *LDLR* gene, it has been reported that statins indirectly induce its gene expression [81,82,83]. *PCSK9* is also regulated by intracellular sterol levels. However, despite low intracellular cholesterol levels inhibiting *PCSK9* gene expression, statins have been shown to induce its expression [81,82,83]. This process causes an attenuation of the lipid-lowering effects of statins. Our results, with respect to the observed statin’s gene expression modulation, demonstrate that the phytocomponents present in PHE have an opposite effect on transcriptional and post-transductional regulation of membrane *LDLR*.

To compensate for the fraction that escapes enterohepatic circulation and is removed by feces, bile salts are de novo synthesized from cholesterol in hepatocytes [84]. This conversion represents the major route for the removal of cholesterol from the body, and it is well known that an increased turnover of bile salts is able to reduce LDL-C [84]. Bile salts regulate their own synthesis in a negative feedback loop by activating *FXR* (Farnesoid X Receptor). As a consequence of the activation, *FXR* reduces the expression of cholesterol-7α-hydroxylase (*CYP7A1*), the first and main rate-controlling enzyme in the pathway of bile salt synthesis [85]. Cell exposure to PHE determined a strong downregulation of *FXR* under all tested concentration ranges (Figure 8). In particular, 60% of its inhibition was already observed at the lowest PHE concentration (1 μg mL^−1^). Since FXR is considered the major regulator of bile salt synthesis, our result suggests that PHE may strongly increase cholesterol turnover.

Increased activity of *CYP7A1* is also associated with the activation of nuclear receptor PPARα (Peroxisome Proliferator-Activated Receptor α), which is principally involved in lipid metabolism regulation. Concerning cholesterol metabolism, PPARα activation intensely downregulates the expression of *HMGCR* [86], promotes cholesterol conversion into bile acids [87], and influences the absorption of cholesterol in the intestinal tract [88]. Consequently, the clinical use of PPARα agonists improves the overall plasmatic lipid profile and reduces cardiovascular risks [89]. In our experimental conditions, cell exposure to PHE increased the gene expression of PPARα. Surprisingly, already at the lowest PHE concentration, an upregulation of the transcript was observed (Figure 8). 

## 3. Materials and Methods

### 3.1. Plant Material and Extract Preparation

*Protium heptaphyllum* oleum-resin was collected in Brazil in April 2019. The bioactive compounds present in the raw material were extracted by Abel Nutraceuticals (Turin, Italy; Extract Batch #P75-2-0) through a patent-pending hydroalcoholic extraction process targeted to increase the concentration of acidic triterpenes present in the resin. The form of the extract provided for this study (which is branded as Hepamyr^®^ by the company in its final commercial powder form) was the solid crude extract of the resin obtained after extraction, purification, and complete solvent removal. For both high-pressure liquid chromatography (HPLC) analysis and biochemical assays, the dried extract was completely resuspended in ethanol (Sigma-Aldrich, Berlin, Germany, EU); meanwhile, diethyl ether (Sigma-Aldrich, Germany) was used for gas-chromatographic (GC) analysis sample preparation.

### 3.2. Chemical Characterization Via HPLC-APCI-HRMS^2^

Semi-untargeted qualitative and quantitative analyses were performed by an HPLC system (Dionex ultimate 3000 HPLC, ThermoFisher Scientific, Waltham, MA, USA) coupled via atmospheric-pressure chemical ionization (APCI) to a high-resolution tandem mass spectrometry instrument (HRMS^2^; LTQ Orbitrap, ThermoFisher Scientific, Waltham, MA, USA). Separation was carried out with Luna C18(2) 150 × 2 mm, 100 Å, 3 µm (Phenomenex, Torrance, CA, USA) using 0.1% (*v*/*v*) formic acid (solvent A) and 0.1% (*v*/*v*) acetonitrile (solvent B) as mobile phases. Chromatographic separation consisted of a solvent ramp from 5% to 100% solvent B for 30 min, followed by column reconditioning of 15 min. The flow rate was set to 0.2 mL min^−1^ and the injection volume to 10 µL. Detection parameters were the following: negative ionization mode; capillary temperature: 250 °C; APCI vaporizer temperature: 450 °C; sheath gas: 35 Arb; auxiliary gas: 15 Arb; discharge needle: 5 kV; the acquisition was carried out in Dependent Scan mode with a mass range from 220 to 1000 *m*/*z*, normalized CE: 35V. Raw data obtained were analyzed with Thermo Xcalibur software (ThermoFisher Scientific, Waltham, MA, USA); molecules were identified using the MetFrag online tool [90]. Quantification of identified acidic triterpenes was performed with a calibration curve of moronic acid (TCI-Europe, Bruxelles, Belgium, EU).

### 3.3. Chemical Characterization via GC-MS and GC-FID

Volatile compounds (VOC) were profiled via GC coupled with a mass spectrometer (MS) and quantified by GC coupled with a flame ionization detector (FID; GCMS-QP2010 SE, Shimadzu, Japan). Qualitative and quantitative analyses were performed using an Rtx-5MS column (30 m; 0.25 mm ID; 0.25 µm film thickness; Restek, Milan, Italy, EU). The injection port was in split mode (split ratio 1:5) for GC-FID analysis or in splitless mode for GC-MS analysis. The temperature of the injection port was kept at 280 °C. The carrier gas was helium, with a constant flow of 1 mL min^−1^. The temperature gradient was as follows: initial temperature 50 °C, then a 3 °C min^−1^ ramp-up to 140 °C and a 12 °C min^−1^ ramp-up to 320 °C. The final temperature was held for 10 min. MS detector conditions were as follows: ionization energy 70 eV, ion source 200 °C, and quadrupole 150 °C; the acquisition was in scan mode (scan range 50–650 *m*/*z*). VOC-targeted identification and quantitation were achieved with a VOC terpene analytical-standard mix (Cannabis Terpene Mix B; CRM40937; Sigma-Aldrich, USA). Meanwhile, the identification was carried out with the NIST database and the FFNSC3 Shimadzu mass spectra library. Identification and quantification of not acid triterpenes (NATs) were carried by GC-MS (TRACE 1310 coupled to TSQ Quantum Ultra, Thermo Scientific, USA). The detection was led in full-mass mode (50–450 *m*/*z* scan range), and the identification was performed by the NIST database. The injection volume was 0.5 µL, and the injector was a PTV in splitless mode at constant temperature (280 °C). The carrier gas was He (1.2 mL min^−1^), and a DB-1 column (30 m × 0.53 mm ID × 5 µm film thickness) was assembled on the GC instrument. The temperature gradient was 150 °C initial temperature, 150 °C at 4 min, 320 °C at 12.5 min, 320 °C at 20 min. For the quantification, the extracted ions were the following: 218, 203, 426, 189 *m*/*z*. Quantification of identified triterpenes was performed with a calibration curve of α- and β-amyrin (TCI-Europe, Belgium).

### 3.4. Determination of Cholesterol Levels

Cholesterol production was evaluated from THLE-3 ATCC^®^ CRL-11233 cells (American Type Culture Collection ATCC, USA). THLE-3 cells were derived from primary normal liver cells by infection with SV40 large T-antigens. Cells were cultured in bronchial epithelial cell growth medium (BEGM; Lonza/Clonetics Corporation, Walkersville, MD, USA) in 5% CO_2_ at 37 °C. The culture flasks used for the experimentation were precoated with a mixture containing 0.01 mg mL^−1^ fibronectin, 0.03 mg mL^−1^ bovine collagen type I, and 0.01 mg mL^−1^ bovine serum albumin dissolved in BEBM medium.

To evaluate cholesterol production by THLE-3 cells, a cholesterol/cholesteryl ester assay kit (Abcam, Cambridge, United Kingdom, EU) was employed. Fluorescent determination was performed following the instructions provided by the manufacturer. Briefly, cells were seeded on 6-well coated plate, and 48 h after seeding, they were treated with the active carboxylate form of lovastatin (CAS 75225-50-2, Enzo Life Sciences, Inc., Farmingdale, NY, USA) at concentrations ranging 0.01–10 μg mL^-1^ in cell medium, PHE (25–200 μg mL^−1^ in cell medium), or simply with BEGM. Cells were treated for 6 and 12 h. After incubation time, the cell monolayer was washed 5 times with phosphate-buffered saline (PBS). Lipid extraction was performed by using 200 μL of 7:11:0.1 (*v*/*v*/*v*) chloroform:isopropanol:NP40 solution. The extracts were then centrifuged for 10 min at 15,000× *g*. Samples were desiccated in a vacuum chamber, and then dried lipids were dissolved in assay buffer, treated with cholesterol reaction mix, and finally incubated for 60 min at 37 °C. Fluorescence was recorded using an F5 FilterMax microplate reader (Molecular Devices, San Jose, CA, USA) set at 535 nm for excitement and at 587 nm for emission.

### 3.5. Evaluation of HMGCR Activity

In order to evaluate the effect of PHE on HMGCR activity, the Colorimetric HMG-CoA Reductase Activity Assay Kit (Abcam, UK) was employed. This assay is based on the consumption of NADPH by the enzyme, which can be consequently measured by the decrease of the absorbance at 340 nm. The assay was performed following the instructions provided by the manufacturer, testing different concentrations of LHA (100–0.01 μg mL^−1^) as a positive control and PHE (10–200 μg mL^−1^). The absorbance was measured using an F5 FilterMax microplate reader (Molecular Devices, San Jose, CA, USA) in Kinect mode for 30 min every 2 min at 37 °C.

### 3.6. Molecular Docking Studies and Validation on HMGCR

#### 3.6.1. Ligand and Protein Preparation

Before in silico studies, both ligands and the protein–ligand complex were prepared as described below. The ligands were optimized for molecular docking using the default setting of the LigPrep Tool implemented in Schrödinger’s software (v2017-1) [91]. All the potential tautomer and stereoisomer combinations were also generated for the biologically relevant pH (pH = 7.0  ±  0.4) using the Epik ionization method [92]. Moreover, energy minimization was done using the integrated OPLS 2005 force field [93]. Regarding the protein, the high-resolution crystal structure of the complex related to the catalytic portion of human HMGCR and lovastatin was downloaded from the Protein Databank (PDB ID: 1HWK) [62,94]. The Protein Preparation Wizard of Schrödinger’s software was employed for protein structure preparation using the condition previously described [95]. Bond orders, hydrogen atoms, and the protonation of the heteroatom states were assigned using the Epik-tool set at biologically relevant pH values (pH = 7.0 ±  2.0). Additionally, the H-bond network was adjusted and optimized. Finally, a restrained energy minimization step (RMSD of the atom displacement for terminating the minimization set at 0.3 Å) was done using the Optimized Potentials for Liquid Simulations (OPLS)-2005 force field on the obtained structure [93].

#### 3.6.2. Molecular Docking

Molecular docking studies were performed using the Glide program [96,97,98]. The receptor grid was prepared by assigning lovastatin as the centroid of the grid box. The 3D structures of the conformers previously generated were then docked into the receptor model using the Standard Precision mode as the scoring function. Five different poses for each ligand conformer were included in the postdocking minimization step, and at least two docking poses for each ligand conformer were generated. The proposed docking procedure was able to redock the crystallized lovastatin within the receptor-binding pocket with RMSD  <  0.51 Å. An IFD application is an accurate and robust method aimed at accounting for both ligand and receptor flexibility [99], and it was performed with the aim of inducing fit docking simulations [91,100] using the Schrödinger software suite [101,102]. The atomic coordinates for c-Kit were downloaded from the Protein Databank (PDB id 1HWK) and submitted to the Protein Preparation Wizard module in Schrödinger in order to add hydrogen, assign partial charges (using the OPLS-2001 force field), confer protonation states, and remove crystal waters. The IFD protocol was carried out as previously reported [103,104]. Briefly, the ligands were docked into the rigid receptor model with scaled-down van der Waals (vdW) radii. The Glide Standard Precision (SP) mode was used for the docking, and the different ligand poses were retained for protein structural refinements [97,98]. In order to include all amino acid (AA) residues within the dimensions of 25 × 25 × 25 Å, the docking boxes were defined from the center of the original ligands [105,106]. The induced-fit protein–ligand complexes were produced using Prime software [105,106]. The previously determined ligand poses, including all residues with at least one atom located within 5.0 Å, were submitted to both side chain and backbone refinements. All poses generated were then hierarchically classified, refined, and further minimized into the active site grid before being finally scored using the proprietary GlideScore function, defined as Equation (1):(1)GScore= 0.065 x vdW + 0.30 x Coul + Lipo + Hbond + Metal + BuryP + RotB + Site
where *vdW* is the energy term for van der Waals interactions; *Coul* is the Coulomb energy; *Lipo* is the lipophilic contact term that refers to potential favorable hydrophobic interactions; *Hbond* is the H-bonding term; *Metal* is the metal-binding term; *BuryP* is a penalty term assigned to buried polar groups; *RotB* is a penalty factor assigned for freezing rotatable bonds; *Site* is a term used to describe favorable polar interactions in the active site. 

Finally, in order to account for both protein–ligand interaction energy and total energy of the system, the *IFD score* was calculated according to Equation (2): (2)IFD score = 1.0 Glide_Gscore + 0.05 Prime_Energy

As a result of this equation, the compounds were ranked for its *IFD score*, considering that the most negative *IFD score* is conferred to the most favorable binding.

### 3.7. Gene Expression via qRT-PCR

Gene expression was evaluated on a HepG2 cancer cell line (American Type Culture Collection ATCC, Manassas, VA, USA). The cells were cultured in RPMI supplemented with 5% (*v*/*v*) FBS, 2 mM L-glutamine, 50 IU mL^−1^ penicillin, and 50 μg mL^−1^ streptomycin, and they were grown in 75 cm^2^ culture flasks under humidity (85–95% H_2_O), temperature (36.5–37.3 °C), and CO_2_ (4.9–5.1%) controlled conditions [22].

When the cells reached about 80% confluence, they were collected using a gamma-irradiated solution containing 0.12% trypsin and 0.02% EDTA in Dulbecco′s phosphate-buffered saline (133.00 mg L^−1^ CaCl_2_, 100.00 mg L^−1^ MgCl_2_, 200.00 mg L^−1^ KCl, 200.00 mg L^−1^ KH_2_PO_4_, 8000 mg L^−1^ NaCl, and 1143.56 mg L^−1^ Na_2_HPO_4_) without phenol red [22]. The cells were seeded in 24-multiwell plates at a density equal to 5 × 10^5^ cells/well. After 24 h, they were treated for 8 h with PHE (1–50 μg mL^−1^ cell medium) in fresh FBS-free RPMI, and total cellular RNA was isolated after their lysis using a commercial kit (GenElute™ Direct mRNA Miniprep Kits, Sigma-Aldrich, USA) [72]. For each sample, the same amount of total RNA (one microgram) was reverse-transcribed using a High-Capacity cDNA Reverse Transcription Kit (Applied Biosystems, USA) and oligo (dT), following the protocol provided by the manufacturer [107]. qMAXSen™ Green qPCR MasterMix (Low ROX™; Canvan, Australia) was employed to analyze the obtained cDNA for quantitative real-time PCR (qRT-PCR) using a QuantStudio™ 3 Real-Time PCR System (Thermofisher, USA) [108]. The qRT-PCR was performed as previously described [72], using the primers reported in Table 3. Finally, relative expression was calculated according to the Pfaffl method [109] using Equation (3):(3)$$Fold\;Change=\;\frac{{\left(E_{GOI}\right)}^{\Delta Ct\;GOI}}{{\left(E_{HKG}\right)}^{\Delta Ct\;HKG}}$$
where *E* is the primer efficiency; *GOI* is the gene of interest; *HKG* is the housekeeping gene; *Ct* is the cycle threshold; ∆*Ct* is the difference between the control average and average *Ct*.

### 3.8. Statistical Analysis

For each experiment, at least three different replicates were performed. The results are expressed as mean values ± standard deviation (SD). Significative differences among the sample were evaluated by the Mann–Whitney test or one-way ANOVA, followed by Tukey’s test. The Mann–Whitney test was employed to analyze data related to cholesterol release and gene expression experiments, in which a direct comparison between control and treatment was required. One-way ANOVA, followed by Tukey’s test, was employed to evaluate what different concentrations of treatment (LHE or PHE) could give in terms of HMGCR activity. Hierarchical cluster analysis, coupled with heat map visualization, was generated using SPPSS software (v24).

## 4. Conclusions

Our results provided information on the functional properties of *Protium heptaphyllum* gum resin extract. Specifically, we have shown that PHE is a rich source of triterpenoic compounds, mainly belonging to the ursane, oleanane, and tirucallane classes, which are limitedly distributed in nature. Moreover, our observations suggest that *Protium heptaphyllum* gum resin has a specific effect on cholesterol metabolism. Therefore, PHE may be a useful alternative to improve dyslipidemia and its clinical complication, such as atherosclerosis and CVDs. For this reason, *Protium heptaphyllum* may become an interesting raw material for the nutraceutical industry in addition to gaining acceptance as a source of health-promoting compounds. Finally, the obtained results may boost the production demand of this resin, improving the income of local rainforest communities. This, together with the application of innovative and sustainable models in the production chain, based on the use of vegetable nontimber forest products, are the base concept of the circular economy [110].

## 5. Patents

A patent related to a part of this work is pending (application number 102020000015598).

## Figures and Tables

**Figure 1 ijms-22-02664-f001:**
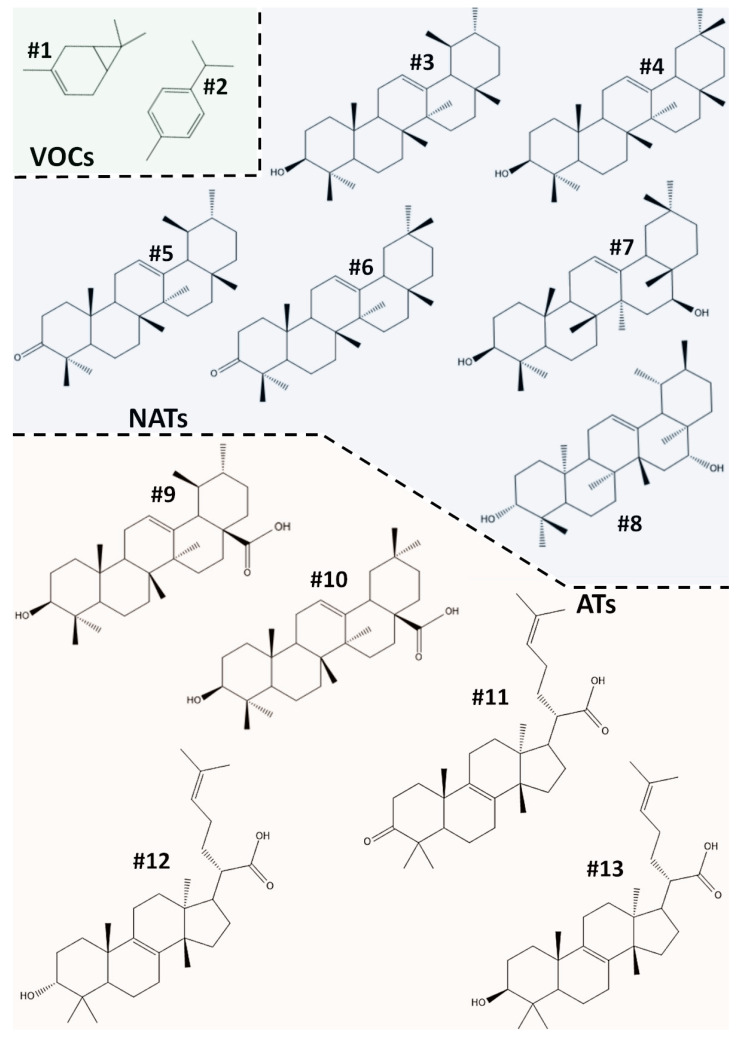
Chemical structures of VOC, NAT, and AT compounds, characterized and quantified in *Protium heptaphyllum* ethanolic extract (PHE). VOCs: volatile organic compounds (mono- and sesquiterpenes); NATs: nonacidic triterpenes; ATs: acidic triterpenes containing a carboxyl group.

**Figure 2 ijms-22-02664-f002:**
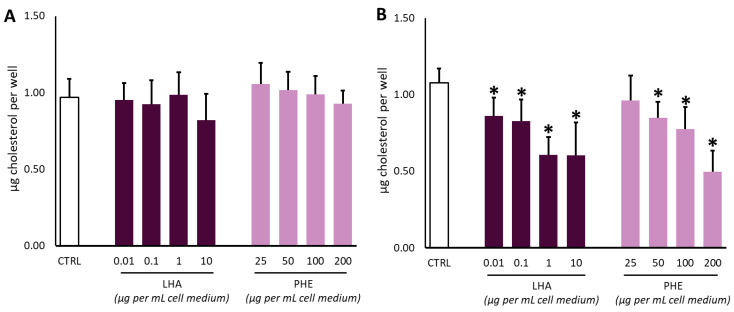
Effect of lovastatin hydroxy acid (LHA) and *Protium heptaphyllum* resin extract (PHE) on cholesterol levels in THLE-3 cells after 6 h (**A**) or 12 h (**B**) of the treatment. Data are expressed as mean ± SD of three different replicates. The symbol “*”, when present, indicates significant (*p* < 0.05) differences between treated and control samples, as calculated by the Mann–Whitney test.

**Figure 3 ijms-22-02664-f003:**
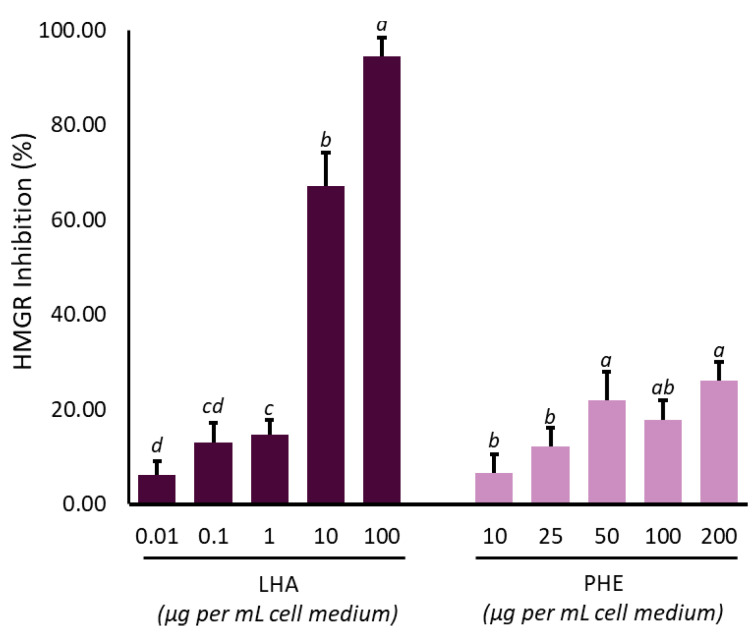
Effect of lovastatin hydroxy acid (LHA) and *Protium heptaphyllum* resin extract (PHE) on HMG-CoA reductase activity. Data are expressed as mean ± SD of three different replicates. Among the two conditions (LHA or PHE), different lowercase letters indicate significant (*p* < 0.05) changes among treatment concentrations, as calculated by one-way ANOVA analysis followed by Tukey’s test.

**Figure 4 ijms-22-02664-f004:**
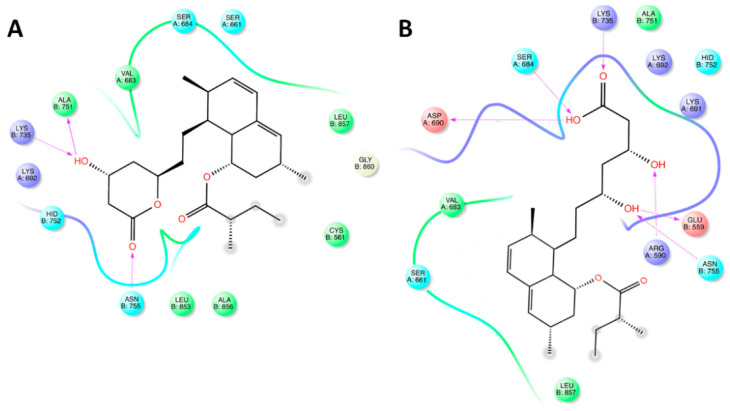
Maps of amino acid (AA) residues involved in the binding of lovastatin (**A**) and lovastatin hydroxy acid (LHA; (**B**)) into 3-hydroxy-3-methyl-glutaryl-coenzyme A reductase (HMGR).

**Figure 5 ijms-22-02664-f005:**
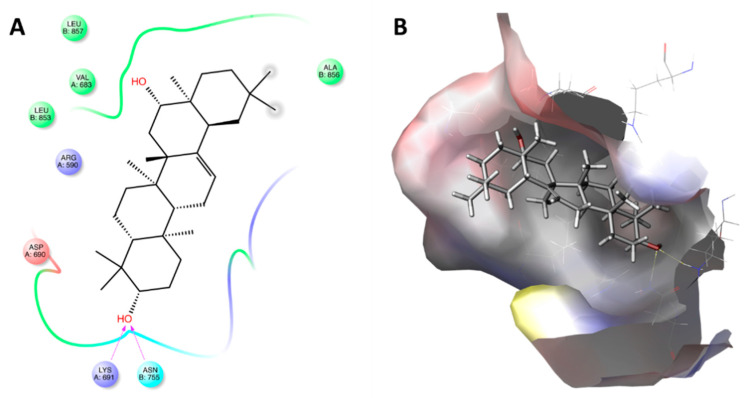
Molecular docking of maniladiol (#8) with the active site of 3-hydroxy-3-methyl-glutaryl-coenzyme A reductase (HMGR). Panel (**A**) shows the amino acid (AA) residues map of the binding. Panel (**B**) shows the 3D docking pose of the molecule into the hydrophobic cavity of HMGCR.

**Figure 6 ijms-22-02664-f006:**
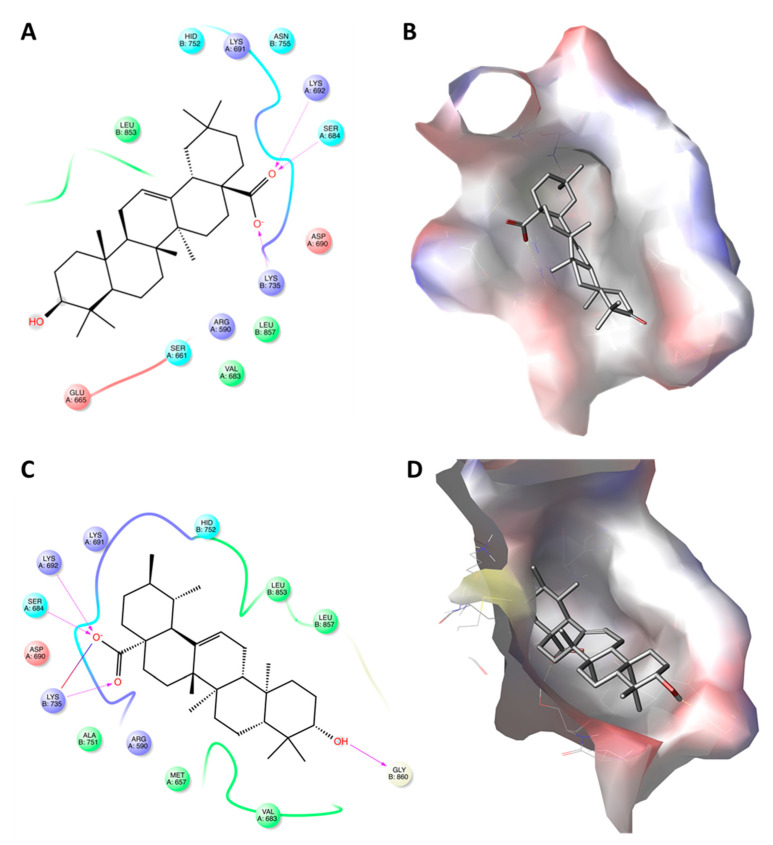
Molecular docking of oleanolic acid (#9) and ursolic acid (#10) with the active site of 3-hydroxy-3-methyl-glutaryl-coenzyme A reductase (HMGR). Panels (**A**,**C**) show the interactions between the amino acid (AA) residues in the active site of HMGR and oleanolic acid and ursolic acid, respectively. Panels (**B**,**D**) show the 3D docking structures of oleanolic acid and ursolic acid, respectively, inserted into the hydrophobic cavity of HMGCR.

**Figure 7 ijms-22-02664-f007:**
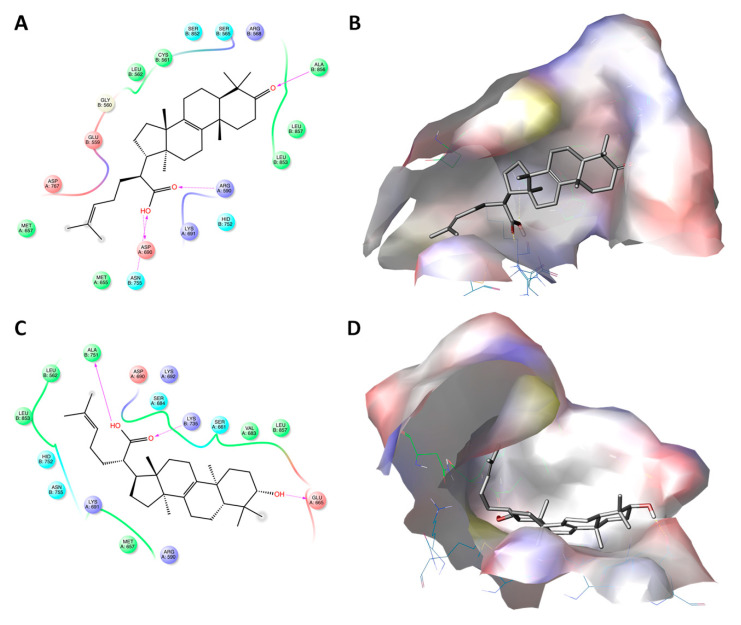
Molecular docking of elemonic acid (#11) and β-elemolic acid (#13) with the active site of 3-hydroxy-3-methyl-glutaryl-coenzyme A reductase (HMGR). Panels (**A**,**C**) show the interactions between the amino acid (AA) residues in the active site of HMGR and elemonic acid and β-elemolic acid, respectively. Panels (**B**,**D**) show the 3D docking structures of elemonic acid and β-elemolic acid, respectively, inserted into the hydrophobic cavity of HMGCR.

**Figure 8 ijms-22-02664-f008:**
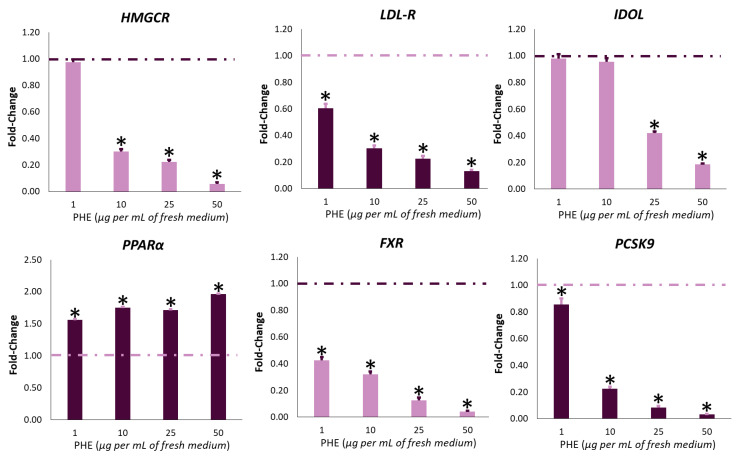
Effect of *Protium heptaphyllum* resin extract (PHE) on gene expression of *HMGCR*, *LDL-R*, *IDOL*, *PPARα*, *FXR*, and *PCSK9* on HepG2 cells. After the seeding, cells were treated for 6 h with PHE. Untreated cells were used as control (dashed line). Bars represent the mean ± SD of three qRT-PCR analyses. Values are expressed as fold change with respect to the gene expression of control cells. The symbol “*”, when present, indicates significant (*p* < 0.05) differences between treated and control samples, as calculated by the Mann–Whitney test. Representative heat map analysis coupled with hierarchical cluster analysis is reported in Figure 9. *HMGCR*: 3-hydroxy-3-methyl-glutaryl-coenzyme A reductase; *FXR*: farnesoid X receptor; *LDLR*: low-density lipoprotein receptor; *IDOL*: inducible degrader of low-density lipoprotein receptor; *PCSK9*: proprotein convertase subtilisin/kexin type 9; *PPARα*: peroxisome proliferator-activated receptor α.

**Figure 9 ijms-22-02664-f009:**
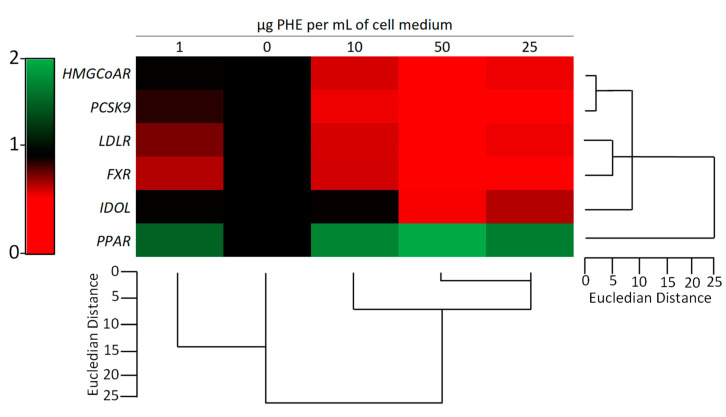
Hierarchical clustering analysis and heatmap visualization of the gene expression of *HMGCR*, *LDL-R*, *IDOL*, *PPARα*, *FXR*, and *PCSK9* on HepG2 cells evaluated via qRT-PCR analysis. For each row, diverse colors indicate differences in gene expression for each gene among the different treatments. *HMGCR*: 3-hydroxy-3-methyl-glutaryl-coenzyme A reductase; *FXR*: farnesoid X receptor; *LDLR*: low-density lipoprotein receptor; *IDOL*: inducible degrader of low-density lipoprotein receptor; *PCSK9*: proprotein convertase subtilisin/kexin type 9; *PPARα*: peroxisome proliferator-activated receptor α.

**Table 1 ijms-22-02664-t001:** Chemical composition of *P. heptaphyllum* gum resin extract. For each compound, CAS ID, MW, chemical formula, name, and percentage composition are reported in the table.

#	CAS ID	MW	Chemical Formula	Class	Name	Composition (%)
1	13466-78-9	136.23	C_10_H_16_	VOCs	Δ3-carene	0.70 ± 0.08%
2	99-87-6	134.22	C_10_H_14_		p-cymene	0.90 ± 0.02%
3	638-95-9	556.9	C_39_H_56_O_2_	NATs	α-amyrin	12.40 ± 1.39%
4	559-70-6	426.7	C_30_H_50_O		β-amyrin	9.50 ± 1.17%
5	638-96-0	424.7	C_30_H_48_O		α-amyron	2.20 ± 0.08%
6	638-97-1	424.7	C_30_H_48_O		β-amyron	1.70 ± 0.61%
7	465-08-7	442.7	C_30_H_50_O_2_		Brein	1.30 ± 0.08%
8	595-17-5	442.7	C_30_H_50_O		Maniladiol	0.90 ± 0.06%
9	508-02-1	456.7	C_30_H_48_O_3_	ATs	Oleanolic acid	0.05 ± 0.01%
10	77-52-1	456.7	C_30_H_48_O_3_		Ursolic acid	0.06 ± 0.01%
11	28282-25-9	454.7	C_30_H_46_O_3_		Elemonic acid	15.60 ± 0.06%
12	28282-27-1	456.7	C_30_H_48_O_3_		α-elemolic acid	10.57 ± 0.76%
13	28282-54-4	456.7	C_30_H_48_O_3_		β-elemolic acid	12.80 ± 1.38%

CAS ID: Chemical Abstracts Service identification number; MW: molecular weight; VOCs: volatile organic compounds (mono- and sesquiterpenes); NATs: nonacidic triterpenes; ATs: acidic triterpenes containing a carboxyl group.

**Table 2 ijms-22-02664-t002:** Molecular docking results of the identified compounds in *Protium heptaphyllum* resin extract (PHE).

#	Compound Name	Docking Score	Prime Energy	IFD Score
	Lovastatin	−6.247	−29124.9	−1462.494
	Lovastatin Hydroxy Acid	−14.667	−29097.0	−1469.500
1	Δ3-carene	−3.220	−29015.2	−1453.979
2	p-cymene	−4.009	−29030.2	−1455.521
3	α-amyrin	−4.637	−29073.5	−1458.313
4	β-amyrin	−4.405	−29079.4	−1458.376
5	α-amyron	−5.196	−29066.6	−1458.528
6	β-amyron	−4.245	−29030.2	−1455.753
7	Brein	−5.307	−29066.2	−1458.666
8	Maniladiol	−5.356	−29151.0	−1462.869
9	Oleanolic acid	−9.835	−29059.2	−1462.795
10	Ursolic acid	−8.008	−29074.3	−1461.721
11	Elemonic acid	−6.786	−29212.4	−1467.408
12	α-elemolic acid	−4.724	−29220.5	−1465.750
13	β-elemolic acid	−6.828	−29166.3	−1465.142

IFD Score: induced fit docking score.

**Table 3 ijms-22-02664-t003:** Genes employed for qRT-PCR analysis. For each gene, the name, acronym, and nucleotide sequences for both Forward (F) and Reverse (R) primers are reported in the table.

Gene Acronym	Sequence (5′−3′)		Accession Number
*HMGCR*	F	TGATTGACCTTTCCAGAGCAAG	NM_000859.2
	R	CTAAAATTGCCATTCCACGAGC	
*FXR*	F	TCTCCTGGGTCGCCTGACT	NM_005123
	R	ACTGCACGTCCCAGATTTCAC	
*LDLR*	F	AGTTGGCTGCGTTAATGTGA	NM_000527.4
	R	TGATGGGTTCATCTGACCAGT	
*IDOL*	F	AAGTTCTTCGTGGAGCCTCA	NM_013262
	R	ACTGAGTTCCACTGCCTGCT	
*PCSK9*	F	GCTGAGCTGCTCCAGTTTCT	NM_174936.3
	R	AATGGCGTAGACACCCTCAC	
*PPARα*	F	CTGGAAGCTTTGGCTTTACG	NM_005036.4
	R	GTTGTGTGACATCCCGACAG	
*βACT*	F	CGGGAAATCGTGCGTGACAT	NM_001101.3
	R	GGACTCCATGCCCAGGAAGG	

*HMGCR*: 3-hydroxy-3-methyl-glutaryl-coenzyme A reductase; *FXR*: farnesoid X receptor; *LDLR*: low-density lipoprotein receptor; *IDOL*: inducible degrader of low-density lipoprotein receptor; *PCSK9*: proprotein convertase subtilisin/kexin type 9; *PPARα*: peroxisome proliferator-activated receptor α; *βACT*: β-actin.

## Data Availability

The data presented in this study are available on request from the corresponding author. The data are not publicly available because a patent related to a part of this work is pending (application number 102020000015598).

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
