# Peer review of "Bioactive Triterpenes of Protium heptaphyllum Gum Resin Extract Display Cholesterol-Lowering Potential"

_ijms, 2021, doi:10.3390/ijms22052664_

Round 1

Reviewer 1 Report

This article describes characterization of Protium heptaphyllum gum resin extract (PHE) for cholesterol lowering effect using hepatocytes and investigates the influence of PHE on some biosynthetic enzymes and the gene expression involved in cholesterol biosynthesis and in relation to the physiological marker of hyperlipidemia.

Preparing the gum resin, and chemical characterization of constituents in the plant material looks sound. Assay method of HMGCoA reductase activity looks fine. Time course of the result figure 2 PHE decreases total cholesterol in hepatocytes is reasonable. The results shown at the table 3 are well supported, yet the active site analyses how the beta-hydroxyl group works on regulating inhibitory activity of pravastin had been carried out already in 20th century during the course of developing variety of derivatives.

Recently, the referee found, there are such articles as follows suitable for reference to quote, Diana S. Gesto, Carlos M. S. Pereira, Nuno M. F. S. Cerqueira and Sérgio F. Sousa, An Atomic-Level Perspective of HMG-CoA-Reductase:Target Enzyme to Treat Hypercholesterolemia; Molecules 2020, 25, 3891,

In-silico studies of HMG-Co A reductase inhibitors present in fruits of Withania coagulans Dunal (Solanaceae), Tooba Lateef*, Sadaf Naeem, Shamim A Qureshi, Tropical Journal of Pharmaceutical Research February 2020; 19 (2): 305-312

Marahatha et al. BMC Complementary Medicine and Therapies (2021) 21:1, https://doi.org/10.1186/s12906-020-03162-5, The experimental parts of molecular docking are well written to maintain reproducibility.

The low inhibitory activity of PHE on HMGCoA reductase in contrast to “the cholesterol lowering effect” derived the authors to the further investigation of influence of terpenoids on not directly involved cholesterol biosynthetic pathway. The results shown as figure 8 and 9, PHE modulates expression of genes not directly corresponding cholesterol biosynthesis as well as related ones, are the key novelty of this article.

The strong point; the multiple approaches to clarify the cholesterol lowering effect of PHE are all solid. Among varietal approach, the referee highly appreciated it of natural products aspect to clarify constituents including assay method of the reductase. The weak point is, yet not a distinguish activity shown by PHE. The result respectively is probably not worthwhile for the interest of readers. But probably it could be hopeful to develop commercially useful material.

After all the referee feels that the result and discussion were reasonably deduced from the experiment. At the same time, the referee consider that this article possibly capture the interest of reader of this journal. 

Author Response

This article describes characterization of Protium heptaphyllum gum resin extract (PHE) for cholesterol lowering effect using hepatocytes and investigates the influence of PHE on some biosynthetic enzymes and the gene expression involved in cholesterol biosynthesis and in relation to the physiological marker of hyperlipidemia. Preparing the gum resin, and chemical characterization of constituents in the plant material looks sound. Assay method of HMGCoA reductase activity looks fine. Time course of the result figure 2 PHE decreases total cholesterol in hepatocytes is reasonable. The results shown at the table 3 are well supported, yet the active site analyses how the beta-hydroxyl group works on regulating inhibitory activity of pravastin had been carried out already in 20th century during the course of developing variety of derivatives. AND The low inhibitory activity of PHE on HMGCoA reductase in contrast to “the cholesterol lowering effect” derived the authors to the further investigation of influence of terpenoids on not directly involved cholesterol biosynthetic pathway. The results shown as figure 8 and 9, PHE modulates expression of genes not directly corresponding cholesterol biosynthesis as well as related ones, are the key novelty of this article. AND The strong point; the multiple approaches to clarify the cholesterol lowering effect of PHE are all solid. Among varietal approach, the referee highly appreciated it of natural products aspect to clarify constituents including assay method of the reductase. AND After all the referee feels that the result and discussion were reasonably deduced from the experiment. At the same time, the referee consider that this article possibly capture the interest of reader of this journal.

We would like to thank the Reviewer for the careful reading of the manuscript and for the punctual and helpful comments/suggestions.

Recently, the referee found, there are such articles as follows suitable for reference to quote,

  1. Diana S. Gesto, Carlos M. S. Pereira, Nuno M. F. S. Cerqueira and Sérgio F. Sousa, An Atomic-Level Perspective of HMG-CoA-Reductase:Target Enzyme to Treat Hypercholesterolemia; Molecules 2020, 25, 3891
  2. In-silico studies of HMG-Co A reductase inhibitors present in fruits of Withania coagulans Dunal (Solanaceae), Tooba Lateef*, Sadaf Naeem, Shamim A Qureshi, Tropical Journal of Pharmaceutical Research February 2020; 19 (2): 305-312
  3. Marahatha et al. BMC Complementary Medicine and Therapies (2021) 21:1, https://doi.org/10.1186/s12906-020-03162-5, The experimental parts of molecular docking are well written to maintain reproducibility.

We thank the reviewer for the suggestion. We introduced the suggested references in our manuscript.

The weak point is, yet not a distinguish activity shown by PHE. The result respectively is probably not worthwhile for the interest of readers. But probably it could be hopeful to develop commercially useful material.

We thank the reviewer for the observation.

In this work we demonstrated a real lowing cholesterol potential of PHE. We are confident that the  obtained data will be important to support our future investigations. We are currently separating the individual components by HPLC in order to better correlate the observed effects with the molecules contained in PHE.

Reviewer 2 Report

It was a pleasure to read the manuscript of Mannino et al. The authors characterize the anti-cholesterol properties of terpene extracts of a resin of the plant species Protium heptaphyllum

The manuscript is well written, clear and the abundantly discussed results confirm the authors' initial assumptions.

Author Response

It was a pleasure to read the manuscript of Mannino et al. The authors characterize the anti-cholesterol properties of terpene extracts of a resin of the plant species Protium heptaphyllum. The manuscript is well written, clear and the abundantly discussed results confirm the authors' initial assumptions.

We would like to thank the Reviewers for the careful reading of the manuscript and for the positive comments.